# From Wristbands to Implants: The Transformative Role of Wearables in Heart Failure Care

**DOI:** 10.3390/healthcare12242572

**Published:** 2024-12-20

**Authors:** Catarina Gregório, João R. Agostinho, Joana Rigueira, Rafael Santos, Fausto J. Pinto, Dulce Brito

**Affiliations:** 1Department of Cardiology, Hospital de Santa Maria (ULSSM), 1649-028 Lisbon, Portugal; joaoragostinho@gmail.com (J.R.A.); joana.rigueira@ulssm.min-saude.pt (J.R.); rafael.santos@ulssm.min-saude.pt (R.S.); faustopin@gmail.com (F.J.P.); dulcebrito59@gmail.com (D.B.); 2Centro Académico de Medicina de Lisboa (CAML), 1649-028 Lisbon, Portugal; 3Cardiovascular Center of the University of Lisbon (CCUL@RISE), 1649-028 Lisbon, Portugal; 4Faculdade de Medicina, Universidade de Lisboa, 1649-028 Lisboa, Portugal

**Keywords:** heart failure, wearable devices, cardiac implantable device, remote patient monitoring, telehealth

## Abstract

Background: Heart failure (HF) management increasingly relies on innovative solutions to enhance monitoring and care. Wearable devices, originally popularized for fitness tracking, show promise in clinical decision-making for HF. This study explores the application and potential for the broader integration of wearable technology in HF management, emphasizing remote monitoring and personalized care. Methods: A comprehensive literature review was performed to assess the role of wearables in HF management, focusing on functionalities like vital sign tracking, patient engagement, and clinical decision support. Clinical outcomes and barriers to adopting wearable technology in HF care were critically analyzed. Results: Wearable devices increasingly track physiological parameters relevant to HF, such as heart rate, physical activity, and sleep. They can identify at-risk patients, promote lifestyle changes, facilitate early diagnosis, and accurately detect arrhythmias that lead to decompensation. Additionally, wearables may assess fluid status, identifying early signs of decompensation to prevent hospitalization and supporting therapeutic adjustments. They also enhance physical activity and optimize cardiac rehabilitation programs, improving patient outcomes. Both wearable and implanted cardiac devices enable continuous, non-invasive monitoring through small devices. However, challenges like data integration, regulatory approval, and reimbursement impede their widespread adoption. Conclusions: Wearable technology can transform HF management through continuous monitoring and early interventions. Collaboration among involved parties is essential to overcome integration challenges and validate most of these devices in clinical practice.

## 1. Introduction

Heart failure (HF) is the leading cause of hospitalizations in developed countries [1]. The natural history of HF is characterized by episodes of acute HF, which may result from the sudden onset or worsening of symptoms in patients with pre-existing chronic HF, or it may occur as the initial presentation of the syndrome (new-onset HF). These acute episodes represent a critical point in the progression of HF syndrome, often requiring immediate medical intervention to prevent further deterioration [2]. Currently, hospitalization for acute HF continues to be associated with unfavorable outcomes, including an in-hospital mortality rate of 4–6%, and rehospitalization and one-year mortality rates ranging from 10 to 30% [3]. Therefore, it is essential to develop technologies with the potential for the early detection of signs and symptoms of HF, thus allowing immediate therapeutic measures that are crucial for preventing hospital admissions and improving patient outcomes [4].

Telemonitoring and wearable technologies have become increasingly important in managing HF [5,6]. Various forms of remote monitoring can be distinguished, ranging from tele-homecare, which involves structured telephone assessments, to sophisticated systems that transmit physiological data such as weight, blood pressure, and heart rate [7]. In patients with cardiac devices, remote monitoring goes further by providing advanced metrics like heart rate variability, detection of arrhythmias, patient activity levels, thoracic fluid accumulation, and pulmonary artery pressure [8].

Wearable technologies are pivotal in enhancing HF care through the continuous, real-time monitoring of patients’ physiological parameters. These non-invasive devices measure common physiological parameters during daily activities and wirelessly transmit the data to healthcare providers or monitoring systems [9,10]. Such devices include smartwatches, rings, wristbands, and patches [11].

The use of wearable devices has expanded significantly in recent years. A population-based study that included adults from the Health Information National Trends Survey in the United States revealed that about one in three Americans uses a wearable device, with nearly 80% of users sharing their data with clinicians to enhance their care [12]. However, this proportion remains lower among patients with cardiovascular disease or those at risk of developing it [12]. These devices could be widely adopted to facilitate continuous assessments for patients and clinicians, extending monitoring and care from hospitals and clinics to the home setting. With the appropriate technology to transmit, organize, and analyze the data, they offer the potential to deliver clinical insights that go beyond the scope of traditional in-person visits [13].

In the current study, we also examined wearable and implantable devices, including implantable cardioverter-defibrillators (ICDs), cardiac resynchronization therapy devices (CRTs), and pulmonary artery pressure monitors. While these devices differ in their applications, they share a common goal of monitoring and managing HF by providing real-time physiological data.

For this literature review, a comprehensive search was conducted using scientific databases, including PubMed and the Cochrane Library. We identified a substantial volume of research, with over eighty-eight publications addressing the use of wearable and implantable devices in heart failure management. We have mainly based the selection for the review on the keywords “heart failure”, “wearable devices”, “cardiac implantable device”, “remote patient monitoring”, and “telehealth”, focusing on review and meta-analyses publications and on pivotal randomized clinical trials.

Our focus in this review was on how these technologies work in concert to enhance HF management, offering an overview of the current evidence and exploring the future potential of wearables in improving patient outcomes. Additionally, we will address the challenges associated with their integration into healthcare systems.

## 2. Wearable Devices and Their Role in HF Management

Wearable technology can monitor various physiological parameters, including heart rate, blood pressure, physical activity, recorded electrocardiograms (ECG), respiratory rate, oxygen saturation, skin temperature, sleep patterns, and activity metrics, through different sensor technologies [12,14] (Figure 1). On the other hand, implantable devices, such as ICDs, CRTs, and CardioMEMS^TM^ HF System, can detect early signs of HF decompensation by measuring thoracic impedance and providing daily pulmonary artery hemodynamic information related to volume overload [15,16]. This approach facilitates personalized care and enables more precise management of HF while encouraging patients to take an active role in their treatment. By offering real-time feedback, wearable devices support patients in adhering to their medication regimens and making necessary lifestyle adjustments [9].

## 3. From At-Risk Patients to HF Diagnosis

The diverse potential applications of wearable devices in HF care should be considered at all stages of the disease. These devices can enhance lifestyle interventions aimed at preventing HF [9]. Wearable technologies, such as step counters, motion sensors, and activity monitors, have been tested across various populations [17]. In the All of Us Research Program, 6042 participants were monitored using Fitbit devices for 5.9 million person-days, revealing that higher daily step counts were associated with a lower risk of developing obesity, sleep apnea, hypertension, and diabetes [18]. In older adults, increased daily step counts corresponded to lower blood pressure (BP), including systolic BP, diastolic BP, mean arterial pressure, and pulse pressure. Participants who achieved a high step count (≥9500 steps/day) and maintained consistent walking patterns experienced the greatest reduction in systolic BP (−1.69 mmHg, 95% confidence interval (CI) −2.2 to −1.18). Conversely, those with moderate step counts (6000 to <9500 steps/day) and regular walking patterns showed the smallest decrease in diastolic BP (−1.067 mmHg, 95% CI −1.379 to −0.755) [19]. A systematic review involving 163,992 participants demonstrated that activity trackers improved physical activity levels (an increase of 1800 additional steps per day), body composition (a reduction of approximately 1 kg in body weight), and fitness (40 additional minutes of walking daily). However, the effects on blood pressure, cholesterol levels, and glycosylated hemoglobin were modest and often not statistically significant [20].

The diagnosis of HF can be challenging, particularly when patients present with nonspecific symptoms or multiple comorbidities that may manifest as dyspnea and fatigue, such as anemia or chronic obstructive pulmonary disease. Therefore, the availability of accurate bedside diagnostic tools can be instrumental in diagnosing patients with symptoms suggestive of HF and can complement the physical examination [21].

In a proof-of-concept study, Shah et al. explored the ability of heart rate variability and entropy measures—specifically the Photoplethysmogram Signal Quality Index (Purity-SQI), standard deviation of beat-to-beat intervals, and standard deviation of accelerometer amplitude—derived from a wristband wearable (Samsung Simband) equipped with a photoplethysmography (PPG) sensor to classify HF in a cohort of hospitalized patients [22]. Participants were instructed to remain still while wearing the device for 5 min in a seated position. Approximately 3 min into the monitoring, they performed five Valsalva maneuvers over the course of 1 min to assess the baroreflex and other heart-related pressure mechanisms. The study included 97 patients, of whom 56% had HF [22]. The wearable-based features alone achieved an overall accuracy of 74% (AUC 0.80). When combined with additional variables, including age, race, history of atrial fibrillation, chronic obstructive pulmonary disease, hypertension, peripheral vascular disease, serum creatinine, systolic blood pressure, diastolic blood pressure, and heart rate, the accuracy improved by 82% (AUC 0.87) [22]. The AUC for the model in HF with reduced ejection fraction (HFrEF) compared to those without HF was 0.92, while for HF with preserved ejection fraction (HFpEF) versus no HF, it was 0.85 [22].

The small sample size limited the ability to derive separate models for HF subcategories based on ejection fraction or ischemic heart disease [22]. These results are particularly relevant in evaluating dyspnea in outpatient or emergency room settings, where time and resources for a comprehensive echocardiogram may be limited.

Specific patterns observed in ECG and PPG signals, such as those recorded by smartwatches and smartphones, can reflect the structural and functional cardiac alterations commonly found in HF patients. On ECG, these abnormalities include poor R-wave progression, prolonged QRS duration, low QRS voltage, prolonged QT interval, and some type of arrhythmias. Such findings are indicative of myocardial remodeling, conduction delays (e.g., left bundle branch block), and arrhythmogenic risks [23]. Regarding PPG signals, abnormalities include reduced signal amplitude, increased pulse wave arrival time, irregular pulse intervals, and blunted waveforms (Figure 1). These findings are associated with changes in cardiac output, vascular compliance, and blood volume dynamics during cardiac cycles, often indicative of the hemodynamic and vascular dysfunction characteristic of HF [24].

These signal-based abnormalities highlight the potential of non-invasive technologies to aid in the detection and management of HF, offering valuable insights into disease progression and therapeutic outcomes.

## 4. Early Detection of Arrhythmias

### 4.1. Atrial Arrhythmias

While it is true that some atrial arrhythmias, such as isolated ectopic beats or short runs, may be benign and unrelated to HF, others, such as atrial fibrillation (AF), are a significant cause of HF decompensation. AF significantly increases the risk of mortality and morbidity associated with stroke and thromboembolism, and it is a common trigger for HF decompensation, which can lead to hospitalization [25]. Data from registries indicate that AF was present in 44% of patients admitted for acute HF and in 37.6% of those with chronic HF [17]. Heart rate variability can be assessed using PPG, a low-cost technology integrated into many devices, enabling the early screening of arrhythmias (Table 1) [26]. In the case of AF, it manifests as varying pulse-to-pulse intervals and pulse morphologies in the PPG signal [23]. PPG sensors can detect AF with a sensitivity and specificity ranging from 91% to 100% compared to electrocardiography [27,28].

Various new smart devices with high sensitivity and specificity for detecting arrhythmic episodes are now available. These include the Alive KardiaBand, Alive KardiaMobile, BodyGuardian, Zio patch, Fitbit Sense, and Apple Watch [29]. The Apple Heart Study was a large, prospective study involving 419,297 participants that assessed the Apple Watch’s ability to detect AF using its photoplethysmography sensor. Over an average follow-up of 117 days, 0.52% (2161 participants) received an irregular pulse notification. Among those notified and monitored with an ECG patch for up to 7 days, 34% were confirmed to have AF. The positive predictive value of the irregular pulse notification was 84% when compared to concurrent ECG recordings. These findings demonstrate the potential of wearable devices for effective AF screening in large populations [30]. In a sub-analysis of this study, among participants who received an irregular pulse notification from the Apple Watch but did not have AF detected on the ECG patch, atrial and ventricular arrhythmias—primarily premature atrial contractions (PACs) and premature ventricular contractions (PVCs)—were identified in 40% of the participants, highlighting the potential of PPG to detect other rhythm abnormalities [31].

When assessing the accuracy of five smart devices [Apple Watch 6—Apple Inc. (Cupertino, CA, USA); Samsung Galaxy Watch 3—Samsung Electronics (Suwon, Republic of Korea); Withings Scanwatch—Withings (Issy-les-Moulineaux, France), Fitbit Sense—Fitbit Inc. (San Francisco, CA, USA), AliveCor KardiaMobile—AliveCor Inc. (Mountain View, CA, USA)] in identifying AF compared to a physician-interpreted 12-lead ECG, the sensitivity and specificity for detecting this arrhythmia varied between 58% and 85% and between 75% and 79%, respectively. The rate of correctly classified tracings was 92% or higher for all the devices evaluated, demonstrating high diagnostic accuracy among the assessed devices when inconclusive tracings were excluded [32].

Patch ECG devices offer a convenient and reliable option for long-term arrhythmia monitoring, providing high-quality ECG recordings with minimal artifacts. They are particularly effective in detecting intermittent arrhythmias, including short episodes of AF, which might be missed by traditional short-term monitoring methods [33]. Their ease of use and extended monitoring capabilities make them a valuable tool for improving arrhythmia detection and patient outcomes.

In addition to detecting AF, other devices have been studied to identify other forms of supraventricular arrhythmias, such as atrial flutter [34], and even bradyarrhythmias [35]. These studies represent an initial step in demonstrating a population-wide strategy for arrhythmia screening using commercially available wearable devices—ranging from smartwatches to single-lead ECG devices attached to smartphones, patches, and smart wristbands—combined with telemedicine.

However, there is a lack of studies focused on the application of these devices in patients with HF and their impact on initiating anticoagulant therapy, as well as on reducing outcomes such as hospitalizations and mortality.

In HF patients who do not currently require a cardiac implantable device or a pacemaker, continuous monitoring for the early detection of tachyarrhythmias could be effectively achieved using an implantable loop recorder (ILR) [36]. ILRs are leadless subcutaneous devices, implanted with a small risk of device-related complications designed for long-term rhythm monitoring [37]. They can operate automatically or be activated by the patient when needed [38]. A small study involving 30 patients with HF and an ejection fraction ≥ 35% in NYHA class II or III, who had not been hospitalized in the last three months, aimed to evaluate the effectiveness of ILRs in early arrhythmia detection for timely intervention. AF was identified in eight patients over an average follow-up of 12 months. Data from the ILR prompted therapeutic adjustments in 13 patients (43.3%). These changes included one pacemaker implant, the initiation of oral anticoagulation in eight patients, and beta-blocker dose adjustments in six patients. Notably, none of the pre-specified ILR events were detected during routine cardiologist visits [8].

In the secondary analysis of the LOOP study [39], participants (older individuals with additional stroke risk factors) who underwent continuous AF screening using an ILR experienced a significant risk reduction in total events for the composite of HFrEF or cardiovascular death, as well as for HFrEF events (hazard ratio (HR), 0.74 [95% CI, 0.56–0.98] and 0.65 [95% CI, 0.44–0.97], respectively), compared to those receiving usual care. This suggests the early identification of AF via ILR can facilitate timely interventions, potentially improving HF management and outcomes in high-risk populations [39].

**Table 1 healthcare-12-02572-t001:** Representative studies of atrial fibrillation detection by wearables in patients with heart failure and their impact on clinical practice. AF—atrial fibrillation; CV—Cardiovascular; ECG—electrocardiogram; HR—hazard ratio; ILR—implantable loop recorder; S—sensitivity (95% confidence interval); Se—specificity (95% confidence interval).

First Author, Year [Ref.]	Study	Device	Study Design	Number of Patients	Primary Outcome	Main Findings
Perez et al., 2019 [30]	Apple Heart Study	Apple Watch	Prospective cohort	419,297	AF detection	34% of the patients with irregular pulse notifications have AF on ECG monitoring. Positive predictive value of 84% for the notifications
Mannhart et al., 2023 [32]	BASEL Wearable Study	Wearable smart devices	Observational	211	AF detection compared with a physician-interpreted 12-lead ECG	The accuracy of all evaluated devices in correctly identifying tracings was at least 92% Apple Watch 6: S 85% (72–94%), Se 75% (67–83%) Samsung Galaxy Watch3: S 85% (72–94%), Se 75% (66–82%) Withings ScanWatch S 58% (42–72%), Se 75% (67–83%) Fitbit Sense S 66% (51–79%), Se 79% (70–86%) AliveCor Kardia Mobile S 79% (64–89%), Se 69% (60–77%)
Xing et al., 2024 [39]	LOOP Study	ILR	Randomized control trial (subset)	6004	Risk reduction in composite outcome for HF event or CV death	Nonsignificant reduction in the primary outcome (HR 0.78). Significant risk reductions in the ILR group for the composite of HFrEF events or cardiovascular death (HR 0.74) and for HFrEF events alone (HR 0.65)

### 4.2. Ventricular Arrhythmias

Regarding the detection of ventricular arrhythmias in patients without an ICD or pacemaker (PMK), the investigators of the VIP-HF study evaluated the incidence of non-sustained ventricular tachycardia (NSVT) in patients with preserved or mid-range ejection fractions [40]. This study involved 113 patients who underwent thorough evaluations, including 24 h Holter monitoring, followed by continuous rhythm monitoring using an ILR for up to two years. Despite a low incidence of ventricular tachycardia (0.6 (95% CI 0.2–3.5) per 100 person-years), the ILR was found to be more effective than Holter monitoring in identifying NSVT, showing an almost 10% higher detection rate (18% on baseline 24 h Holter monitoring and 27% with ILR). Interestingly, the study found that ventricular arrhythmias were not linked to increased hospitalization or mortality risk. Nevertheless, the ILR also demonstrated significant capability in uncovering bradyarrhythmias [40]. Cicogna et al. [36] considered that patients with HF who do not require an ICD or PMK are suitable candidates for ILR implantation. In particular, patients that were not on anticoagulants and without contraindications, and those with conduction delays or myocardial scars, could benefit significantly. The authors proposed that patients experiencing palpitations within the past six months may have a higher chance of detecting relevant arrhythmias.

Despite the evidence supporting the detection of arrhythmic events, more studies are needed to assess the prognostic impact of this monitoring strategy. The effective management of implantable cardiac monitoring systems should occur in HF clinics, with timely evaluations of transmissions to enable swift therapeutic responses.

## 5. Early Detection of Decompensation in HF Patients

Non-invasive telemonitoring involves tracking patients’ health status using electronic devices at home, and it consists of collecting and transmitting weights, vital signs, and symptoms via telephone or wireless methods [41]. However, the benefits of telemonitoring are not consistent across all studies [42] and require significant time and resources from both patients and clinicians [41,43].

### 5.1. Remote Hemodynamic Monitoring Devices

Alternatively, hemodynamic monitoring provides valuable insights for HF patients at high risk of decompensation, requiring frequent interventions to avoid congestion (Table 2).

The CardioMEMS HF System is an advanced device designed to improve HF management. It features a small, wireless sensor implanted in the pulmonary artery via a minimally invasive procedure. This sensor continuously measures pulmonary artery pressure, a critical indicator of worsening HF, often detectable before symptoms arise [44].

In the CHAMPION trial (CardioMEMS Heart Sensor Allows Monitoring of Pressure to Improve Outcomes in NYHA Class III HF Patients), pulmonary artery pressure monitoring with CardioMEMS demonstrated a 28% reduction in HF hospitalizations, with a hazard ratio of 0.72 (95% CI: 0.59–0.88; *p* = 0.0013) [44]. This benefit was sustained up to 12 months post-implantation [45] and was particularly notable in patients receiving ACE inhibitors or angiotensin receptor blockers alongside beta-blockers, who experienced a 43% reduction in HF hospitalization rates (HR: 0.57; 95% CI: 0.45–0.74; *p* < 0.0001).

**Table 2 healthcare-12-02572-t002:** Key studies of remote hemodynamic monitoring devices in patients with heart failure. HF—heart failure; KCCQ—Kansas City Cardiomyopathy Questionnaire.

First Author, Year [Ref.]	Study	Device	Device	Study Design	Number of Patients	Primary Outcome	Main Findings
Givertz et al., 2017 [44]	CHAMPION Trial	Implantable pulmonary artery pressure monitoring devices	CardioMEMS	Randomized control trial	550 patients (NYHA III)	HF hospitalization and mortality rates at 6 months	Reduction in HF hospitalization; no decrease in all-cause mortality was observed
Zile et al., 2022 [46]	GUIDE-HF Trial	Implantable pulmonary artery pressure monitoring devices	CardioMEMS	Randomized control trial	1000 patients (NYHA II–IV, prior HF hospitalization within 12 months or elevated BNP)	All-cause mortality and total HF events at 12 months	Reduction in the composite endpoint of HF hospitalizations, all-cause mortality and urgent heart failure visits
Brugts et al., 2023 [47]	MONITOR-HF Trial	Implantable pulmonary artery pressure monitoring devices	CardioMEMS	Randomized control trial	348 patients (NYHA III and ≥1 HF hospitalization or urgent visit requiring intravenous diuretics)	Mean difference in the KCCQ overall summary score at 12 months	Improvement in quality of life (by KCCQ) and reduction in HF hospitalization
Guichard et al., 2024 [48]	PROACTIVE-HF Trial	Implantable pulmonary artery pressure monitoring devices	Cordella Endotronix	Randomized control trial	456 patients (NYHA III)	HF hospitalization or all-cause mortality rate (Primary effectiveness endpoint at 6 months)	Lower incidence of HF hospitalizations or all-cause mortality at 6 months Freedom from device and system-related complication and from pressure sensor failure

The study focused on patients with HFrEF, New York Heart Association (NYHA) class III symptoms, and a recent history of hospitalization for HF. These patients were instructed to take daily pulmonary artery pressure measurements using the CardioMEMS pillow, which transmitted the data to a central database. Standardized guidelines were then applied to manage patients based on daily pressure trends. Notably, these benefits extend to patients in NYHA classes II-IV and even to subgroups with HFpEF [46].

Furthermore, a significant improvement in quality of life was observed, as measured by the Kansas City Cardiomyopathy Questionnaire (KCCQ), with an increase of +7.05 points in the CardioMEMS group (*p* = 0.0014) compared to −0.08 points in the standard care group (*p* = 0.97) [47]. Another promising device, the Cordella pulmonary artery (PA) sensor (Endotronix), supports HF management through PA pressure-guided therapy, combined with vital sign data to optimize guideline-directed medical therapy and enhance HF outcomes remotely. It demonstrated a lower incidence of HF hospitalizations or all-cause mortality (0.15, 95% CI: 0.12–0.20) compared to previous studies (0.15 vs. 0.43, *p* < 0.0001) [48]. Additionally, implantable left atrial pressure monitoring devices, positioned as an interatrial leadless sensor, showed a strong correlation with invasive measurements of pulmonary capillary wedge pressure, and significant improvements in NYHA functional class and 6 min walk test distance [49].

### 5.2. Use of ICD, CRT, and PMK

Incorporating implantable devices, such as ICD, CRT devices, and PMK, has become foundational in HF management. A PMK is indicated for HF patients with symptomatic bradyarrhythmias due to atrioventricular block or sick sinus syndrome, conditions that lead to reduced cardiac output. An ICD is recommended for the primary prevention of sudden cardiac death in HFrEF patients (≤35%) who remain symptomatic despite optimal medical therapy and have an expected survival of at least one year. Secondary prevention indications include survivors of ventricular arrhythmias or cardiac arrest unrelated to reversible causes. A CRT device, either CRT-P (pacing) or CRT-D (combined with defibrillation), is strongly indicated for patients with HFrEF (≤35%), sinus rhythm, NYHA class II-IV symptoms, and evidence of electrical desynchrony (e.g., left bundle branch block with QRS duration ≥150 ms) [50].

These devices provide therapeutic benefits and advanced monitoring capabilities, helping prevent sudden cardiac death and improve cardiac function while offering continuous, real-time data on critical clinical parameters [51], including thoracic impedance, heart rate variability, heart sound intensity, and patient activity.

Several studies have explored diagnostic algorithms based on cardiac implantable electronic devices, utilizing various monitoring parameters.

The Multigene (Multisensor Chronic Evaluation in Ambulatory HF Patients) trial involved 900 patients with CRT-D devices (with bipolar right atrial, right ventricular, and left ventricular leads), aiming to develop and validate the HeartLogic algorithm for HF risk stratification. This study collected data on heart rate, heart sounds, respiratory rate, relative tidal volume, thoracic impedance, and physical activity through an accelerometer. The HeartLogic algorithm achieved a sensitivity of 70% (95% CI, 55.4–82.1) and a specificity of 85.7% for detecting HF exacerbations, with a median time of 34 days (IQR: 19 to 66.3 days) between alert activation and the onset of an HF event [52]. When applied to a population of patients with ICDs, the same algorithm showed similar results [53].

Analysis of all-cause hospitalizations based on alert status revealed 1.35 events per patient-year during alert periods, compared to 0.50 events per patient-year when not in alert, indicating a hospitalization rate that was 2.7 times higher during alert periods (*p* < 0.001) [53].

A meta-analysis of the TRUST, ECOST, and IN-TIME trials evaluated Biotronik Home Monitoring (HM) in ICD patients, showing improved outcomes [54]. Home Monitoring reduced all-cause mortality by 1.9% (*p* = 0.037) with a risk ratio of 0.62 and significantly lowered the combined endpoint of all-cause mortality or HF hospitalization by 5.6% (*p* = 0.007). This benefit was attributed mainly to the prevention of HF exacerbations [54].

### 5.3. Advances in Wearable Devices

Patients recently discharged with acute decompensated HF are at a high risk of clinical decline and frequent hospital readmissions, needing careful monitoring and intervention [55]. Several studies have investigated the impact of wearable devices on predicting HF decompensation.

The Multisensor Non-invasive Remote Monitoring for Prediction of HF Exacerbation (LINK-HF) study employs a machine learning algorithm to predict HF readmissions. Using a wearable sensor (Vital Connect) placed on the chest, which collects ECG waveforms, accelerometry data, skin impedance, temperature, activity, and posture, the study successfully detected precursors to hospitalization for HF exacerbation with a sensitivity ranging from 76% to 88% and a specificity of 85%. Alerts indicating a likely HF exacerbation can be generated at a median of 6.5 days before admission [56], enabling clinicians to adjust medical therapy and potentially prevent hospitalization. In patients recently hospitalized for HF, the Zoll HFMS system—a small patch that collects various physiological parameters, including thoracic fluid—resulted in a 38% relative risk reduction in HF readmissions. The number needed to treat is 14.3 patients to prevent one readmission due to HF. Additionally, the Zoll HFMS improved quality of life, as measured by KCCQ-12, by an average of 12 points compared to the control (*p* = 0.004). This intervention may involve changes in medication, adjustments to lifestyle or diet, or follow-up visits, as determined by investigators based on data from the wearable device [57].

Wearable vests that measure intrathoracic impedance have demonstrated a good correlation with fluid status. The Remote Dielectric Sensing (ReDS, Sensible Medical) technology measures the dielectric properties of tissues and assesses patients’ volume status post-hospital discharge, guiding adjustments in diuretic therapy. In patients with HFrEF and HFpEF, the use of ReDS was associated with a lower rate of 30-day cardiovascular readmissions and a trend toward lower all-cause readmissions compared to patients without a ReDS assessment [58].

Advances in machine learning are improving HF monitoring. The SCALE-HF 1 trial (Surveillance and Alert-Based Multiparameter Monitoring to Reduce Worsening HF Events) is a multicenter study assessing the Bodyport cardiac scale’s ability to predict HF events. This study included 329 patients with chronic HF and recent decompensation, who took daily home measurements by standing on the scale for 20–30 s. The cardiac scale uses a composite algorithm with ECG, ballistocardiogram (BCG), and impedance plethysmography signals to create a congestion index, generating alerts when a set threshold is exceeded [59]. This index successfully predicted 70% of HF events (48 of 69) at 2.58 alerts per participant-year, outperforming the standard weight rule (weight gain of >3 lb in one day or >5 lb in seven days), which only predicted 35% of events at a higher alert rate (4.18 alerts per participant-year). The congestion index showed significantly higher sensitivity and a lower alert rate, proving more effective than traditional weight-based monitoring [59].

## 6. Treatment Adjustments and Prognosis

There is a limited number of studies focusing on enhancing guideline-medical therapy and monitoring response to treatment in patients with HF, in addition to adjusting diuretic therapy. The RATE-AF trial utilized a wrist-worn wearable linked to a smartphone for 20 weeks to compare heart rates in older patients with permanent AF and HF who were randomized to receive either digoxin or beta-blockers [60]. Heart rates were similar between the two groups, and no differences were observed after accounting for physical activity or among patients with high activity levels. Furthermore, the wearable sensor data appeared comparable to conventional trial outcomes (including ECG heart rate and the 6 min walk test) for predicting NYHA functional class at the end of the trial (wearable score 0.56 [95% CI 0.41 to 0.70] versus conventional parameters 0.55 [95% CI 0.41 to 0.68]; *p* = 0.88 for comparison) [60].

The NEAT-HFpEF (Nitrate’s Effect on Activity Tolerance in HF with Preserved Ejection Fraction) trial evaluated the impact of isosorbide mononitrate (ISMN) on exercise tolerance in patients with HFpEF [61]. The hypothesis was that nitrates, by reducing preload, could enhance exercise capacity. The study employed a double-blind, crossover design in which patients were randomized to receive varying doses of ISMN or a placebo, with activity levels monitored using two hip-worn tri-axial accelerometers. In this population, nitrate use was associated with lower activity levels in a dose-dependent manner [62].

In the AWAKE-HF study, patients with HFrEF receiving sacubitril/valsartan showed no differences in mean activity levels, sleep, or quality of life—as assessed by the KCCQ-23—compared to those receiving enalapril, when evaluated using wearable digital sensors with accelerometers [63].

Iliodromitis et al. investigated the ability of the wearable cardioverter-defibrillator (WCD—LifeVest), equipped with an integrated accelerometer, to assess physical activity in patients with newly diagnosed severely reduced HF (LVEF ≤ 35%) of either ischemic or non-ischemic origin [64]. The study demonstrated a significant increase in the number of daily steps between the first two weeks and the last two weeks of an average follow-up period of 77.3 ± 44.6 days (mean steps in the first 2 weeks: 4952.6 ± 3052.7 vs. mean steps in the last 2 weeks: 6119.6 ± 3776.2, *p* < 0.001). This increase was attributed to improved physical capacity and enhanced patient confidence in exercising after the index event. Previous studies have already established that the WCD accelerometer is a reliable tool for assessing physical activity, with results comparable to the 6 min walk test, while also providing the advantage of applicability in everyday, out-of-hospital settings [65]. Moreover, in patients with high adherence to WCD therapy, early physician intervention can be facilitated by identifying a gradual reduction in physical activity, allowing for timely adjustments to medical therapy and potentially preventing unnecessary hospitalizations.

## 7. Improving Exercise and Cardiac Rehabilitation

Exercise training, which reduces hospitalizations and improves quality of life, is a Class I recommendation by the European Society of Cardiology [66]. However, limited access to facilities, trained staff, and low patient adherence—particularly among frail patients—are common barriers to exercise training in HF [67]. Wearable devices have been suggested as a means to overcome these challenges. The Smart HEART clinical trial evaluated whether a 3-month home exercise program, enhanced by digital health interventions (multi-component coaching, the Movn smartphone app, and a wearable activity tracker), provided greater access and improved patient-centered outcomes compared to a center-based program [63]. In a population of 1653 patients, including 258 participants in the study group (14% of whom had HF), significant improvements were observed in the 6 min walk test (mean difference [MD] −29 m; 95% CI, 10 to 49; *p* < 0.01) and low-density lipoprotein cholesterol (MD −11 mg/dL; 95% CI, −17 to −5; *p* < 0.01). Additionally, the proportion of patients who reported smoking decreased [68].

A systematic review on the safety and efficacy of digital therapeutics-based cardiac rehabilitation for HF patients, which included 1119 participants mostly classified as NYHA Class I or II with an average left ventricular ejection fraction of 35.5%, showed that home-based rehabilitation appeared safe, with no exacerbations of HF leading to unscheduled hospitalizations. The program was feasible, demonstrating an over 85% adherence and high satisfaction rates [69].

Most studies reported improved quality of life and functional capacity [69]. In the TELEREH-HF trial conducted by Piotrowicz et al. [70], patients with HFrEF were monitored using a mini-ECG device, blood pressure monitor, and body weight scale. The intervention increased peak oxygen consumption (VO2 peak) by 0.95 mL/kg/min (95% CI, 0.65–1.26) compared to 0.00 mL/kg/min (95% CI, −0.31–0.30) in the usual care group (*p* < 0.001). Additionally, the 6 min walk distance improved by 30.0 m (95% CI, 24.7–35.3) versus 20.7 m (95% CI, 15.4–26.0) in the usual care group (*p* = 0.01) [70]. These benefits also extend to NYHA Class III HF [71], and ischemic cardiomyopathy patients [72]. However, the impact of telerehabilitation using wearables on major endpoints such as HF hospitalization, cardiovascular mortality, and all-cause mortality is not significant [70,73].

## 8. New Wearables and Future Perspectives

With the rapid growth of wearable technologies in everyday life and healthcare, many devices are still under development, and only a limited number have been validated and approved for commercialization. Sweat analysis devices [74] appear promising as they non-invasively and in real-time measure electrolytes, pH levels, and other metabolic indicators that correlate with fluid balance, kidney function, and overall cardiovascular health. By promoting adequate fluid balance, these devices may allow for the early detection of worsening HF and guide more personalized treatment plans.

The prospective trial (LacS-001) evaluated sweat lactate dynamics during exercise through a small, non-invasive device in patients with NYHA class I or II HF, identifying a strong correlation between lactate threshold in sweat and ventilatory threshold (r  =  0.651, 95% CI 0.391–0.815, *p*  <  0.001). This sensor enables the determination of the anaerobic threshold, optimizing exercise intensity and exercise training programs for cardiac rehabilitation, thus overcoming some limitations of exercise tests combined with respiratory gas analysis [74].

A new sensor for implantation in the vena cava has been developed as an additional tool for predicting congestion. In the FUTURE-HF2 trial, it demonstrated safety and excellent agreement with concurrent Computed Tomography scans (R^2^ = 0.99, mean absolute error = 11.15 mm^2^) [75].

Speech measures using a smartphone application for processing and analysis (HearO) appear useful for identifying voice alterations reflective of changes in acute decompensated HF clinical status, with 94% of the discharge recordings (dry) being distinctly different from their respective baseline (wet) recordings [76].

Endothelial dysfunction and issues in systemic microcirculation are key factors in the development of HF. Monitoring subtle changes in the microvascular system, along with cardiac imaging and biomarkers, may enhance the prediction and prevention of HF [72]. Retinal imaging offers a non-invasive way to observe the microvasculature of the retina. Changes in retinal vessels and retinopathy may result from systemic inflammation, endothelial dysfunction, and cardiovascular diseases. Growing evidence suggests retinal imaging could help detect microvascular issues associated with left ventricular remodeling and HF [77,78]. Smartphone-based retinal cameras and the integration of artificial intelligence–based assessment techniques could serve as an additional tool to prevent HF and stratify risk for those at risk of developing HF [77]. Chandra et al. demonstrated that decreased central retinal arteriolar equivalent, increased central retinal venular equivalent, and arteriolar-to-venular ratio were significantly linearly associated with incident HF after adjusting for age, gender, and race during a 16-year follow-up [78].

Regarding identifying and managing risk factors and comorbidities in patients with HF, Lee et al. investigated the potential of a sock fitted with a BCG sensor to distinguish between healthy individuals and patients with diabetes. This method showed a good correlation in determining heart rate when compared to ECG (correlation coefficient of 0.99 [95% CI 0.99–1.00]), and pressure measurements on the metatarsal area of the foot indicated that the sock could identify patients with diabetes, as well as those with diabetes and poor circulation [79]. In another study, using the wearable sensor SmartSock proved useful in quantifying ankle edema [80].

Non-invasive techniques using accelerometer sensors to measure the vibrations of the jugular vein and the carotid artery, as well as non-invasive, non-contact Frequency Modulated Continuous Wave (FMCW) radar technology for JVP measurements, are developing methods that appear promising and may benefit clinicians and specialists involved in HF patient care, potentially influencing decisions regarding rehospitalization or readmission [81].

The non-invasive determination of serum potassium levels through T-wave morphology analysis in 48-hour Holter recordings in end-stage renal disease patients undergoing hemodialysis (HD) demonstrated high accuracy during HD and in the post-HD period [82]. When applied to HF, this tool could assist in therapy titration and monitoring, eliminating the need for multiple blood draws for the same purpose.

## 9. Challenges to Implementing Wearable Data in Everyday Clinical Care

The implementation of wearable devices in clinical practice for managing HF presents several barriers (Figure 2), including which specific parameters should be monitored, how to integrate and manage the massive amounts of data within current clinical systems, and how to determine appropriate responses to emerging data patterns and alerts.

One of the primary issues is the overwhelming amount of data generated by these devices and the quality of this information. To make appropriate clinical decisions, we need to generate high-quality evidence. Integrating this vast amount of information into existing electronic health record systems and the clinical workflow can be challenging. Clinicians are concerned that the influx of wearable data could increase their workload, especially if the data are not well-organized or presented in an easy way to interpret. There is a need for systems that can filter and prioritize data to provide clear, actionable insights without adding complexity to the clinician’s daily routine [83].

Ensuring that data from wearables can be securely and efficiently stored in personal and electronic health records is technically demanding and costly [83]. Additionally, while wearables provide real-time monitoring, the precision of the information can vary depending on the device used. Inconsistent or inaccurate data can lead to false alarms or missed early signs of HF deterioration, undermining the clinical utility of these tools [83]. Wearable technology must meet the rigorous standards required for clinical decision-making to ensure successful implementation.

Patient adherence is another significant obstacle. Wearable devices require consistent use and patient engagement. However, many patients, particularly those older or with multiple comorbidities, may find it difficult to comply with the regular use of these devices. Discomfort, technical difficulties, or a lack of understanding of the importance of these tools can reduce adherence, limiting their effectiveness in managing HF. In the prospective SafeHeart trial, long-term adherence to a wrist-worn accelerometer was evaluated for the continuous day-to-day collection of physical activity and behavior data to potentially predict worsening health [84]. During the 6-month study, patients were instructed to wear the wrist-worn accelerometer day and night, resulting in a median long-term adherence of 88.2% (Interquartile Range 74.6–96.5%); however, only 83 participants (28%) had optimal adherence. Conversely, the study noted lower adherence among younger patients [84]. Text messages or phone calls may promote adherence to wearables [85], gamification, and financial incentives [86].

Health equity and access also play a critical role in the implementation of wearables. The cost of these devices and the digital divide may prevent some patients—especially those from lower socioeconomic backgrounds or with lower digital literacy—from accessing this technology. Without addressing these disparities, the integration of wearables into clinical practice risks exacerbating existing inequalities in healthcare. A survey conducted on the use of wearable devices in the United States from 2020 to 2022 found that ownership tends to be lower among older adults, individuals with lower incomes and education levels, and those living in rural areas [83]. Compared to younger individuals aged 18–24, those 65 and older are significantly less likely to own wearable devices, with an odds ratio (OR) of 0.18. Gender also plays a role, as women have slightly higher odds of owning wearables than men (OR 1.10). Health insurance status influences ownership; those with private insurance are more likely to own wearables (OR 1.28), while uninsured individuals are less likely (OR 0.41) [87].

Finally, the cost of wearable devices and the lack of clear reimbursement pathways remain significant barriers to widespread adoption. Many healthcare systems and insurers hesitate to cover these technologies, limiting their broader use in clinical settings. A greater number of cost-effectiveness studies on these devices in HF management are needed to promote their widespread adoption.

## 10. Conclusions

Integrating wearable devices and implantable sensors in HF management represents a significant transformative step forward in personalized healthcare. These technologies offer real-time, actionable insights into patients’ physiological status, enabling the early detection of decompensation and more precise interventions. While promising, widespread adoption requires overcoming challenges related to data integration, patient adherence, cost, and accessibility. Future advancements in cost-effectiveness studies, health equity, and technological precision will be crucial in unlocking the full potential of these innovations, ultimately improving outcomes and quality of life for HF patients.

## Figures and Tables

**Figure 1 healthcare-12-02572-f001:**
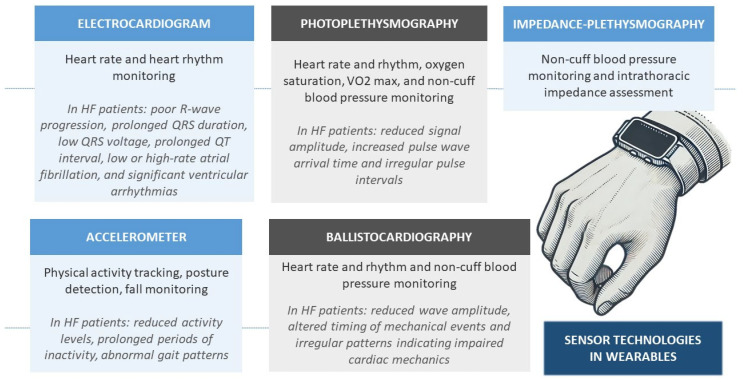
Overview of some sensor technologies in wearables for heart failure management.

**Figure 2 healthcare-12-02572-f002:**
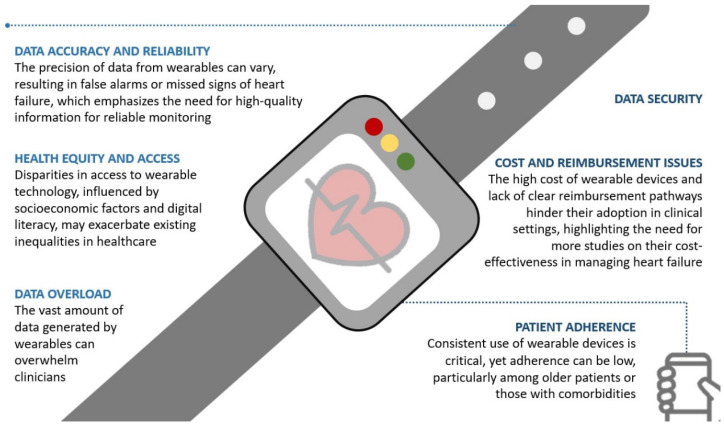
Barriers to the effective implementation of wearable devices in clinical practice for heart failure care.

## Data Availability

The data generated or analyzed during this study are available from the corresponding author upon reasonable request.

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
