# Peer review of "From Wristbands to Implants: The Transformative Role of Wearables in Heart Failure Care"

_healthcare, 2024, doi:10.3390/healthcare12242572_

Round 1
Reviewer 1 Report
Comments and Suggestions for Authors
This is an overview paper showing current status of wearables related to heart failure detection and prediction as well as arrhythmia detection and prediction. It addresses the most recent devices and studies. It is a useful reading for anybody in the filed who want to catch up with recent developments.
Major comments:
1. I think the distinction between HF and arrythmias need to be made because atrial arrhythmias are quite benign so their detection does not have much to do with HF. Therefore, it seems to me this subsection title is not appropriate because you want to detect either atrial arrhythmia, of pathological arrhythmias or HF: "Early Detection of Arrhythmias in HF Patients".
2. Perhaps define what heart failure is and distinguish it from arrhythmias.
Minor comments:
1. This needs a reference: Atrial fibrillation (AF) significantly increases the risk of mortality and morbidity associated with stroke and thromboembolism, and it is a common trigger for HF decompensation, which can lead to hospitalization.
2. Patch ECG devices are a viable option but not discussed as a separate entity. Example: https://www.mdpi.com/1424-8220/20/6/1695 . Maybe some of them are mentioned but they maybe can be a separate topic because they are not the same as a watch. No need to make any changes regarding this comment. It is just a suggestion.
Author Response
We appreciate your comments and suggestions, which have helped enrich our manuscript.
Major comments:
- I think the distinction between HF and arrythmias need to be made because atrial arrhythmias are quite benign so their detection does not have much to do with HF. Therefore, it seems to me this subsection title is not appropriate because you want to detect either atrial arrhythmia, of pathological arrhythmias or HF: "Early Detection of Arrhythmias in HF Patients".
- Perhaps define what heart failure is and distinguish it from arrhythmias.
We appreciate your comment. The authors of the manuscript have clarified that not all atrial arrhythmias contribute to HF decompensation or impact its prognosis. Addressing the topic of "arrhythmias" within the context of HF remains relevant to us; however, studies specifically focused on arrhythmia detection in patients with HF or other medical conditions are still limited. This limitation is acknowledged by the authors. Minor adjustments have been made to the title and the body of the text. We hope these changes align with your suggestions.
Minor comments:
- This needs a reference: Atrial fibrillation (AF) significantly increases the risk of mortality and morbidity associated with stroke and thromboembolism, and it is a common trigger for HF decompensation, which can lead to hospitalization.
Thank you very much for your comment. The reference has been added and corresponds to the article published in JACC Heart Failure - Carlisle MA, Fudim M, DeVore AD, Piccini JP. Heart Failure and Atrial Fibrillation, Like Fire and Fury. JACC Heart Fail. 2019 Jun;7(6):447-456. doi: 10.1016/j.jchf.2019.03.005. PMID: 31146871.
- Patch ECG devices are a viable option but not discussed as a separate entity. Example: https://www.mdpi.com/1424-8220/20/6/1695 . Maybe some of them are mentioned but they maybe can be a separate topic because they are not the same as a watch. No need to make any changes regarding this comment. It is just a suggestion.
Thank you very much for your valuable comment. We have included a brief note highlighting the importance of patch ECG devices and their utility in monitoring atrial arrhythmias. However, we decided not to create a separate section, as we believe this would make the manuscript overly extensive and potentially less accessible for the reader.
Reviewer 2 Report
Comments and Suggestions for Authors
please check comment in the file

Author Response
- The manuscript presents wearable devices (including wristbands and implants) to monitor biomarkers to predict heart failure of patients.
- The reviewers have the following comments:
2.1. In the introduction, the authors mentioned....we examined wearable and implantable devices, including implantable cardioverter-defibrillators (ICDs), cardiac resynchronization therapy devices (CRTs), and pulmonary artery pressure monitors. While these devices differ in their applications, they share a common goal of monitoring and managing HF by providing realtime physiological data. Our focus was on how these technologies work in concert to enhance HF management, offering an overview of the current evidence and exploring the future potential of wearables to improve patient outcomes. Additionally, we will address the challenges associated with their integration into healthcare systems. The reviewer believes that this is a very good way to pose the problem for a review manuscript. However, the rest of the manuscript does not focus on analyzing the problems raised.
2.2. The author should focus on analyzing the technologies of each of the above devices.
2.3. Although it is a review manuscript, the author should explain more clearly with figures about the problems and solution. For example, detecting HF by using ECG/PPG should describe the ECG/PPG signal when HF is present.
Thank you for your comment and suggestions. The authors have clarified and defined each device mentioned, specifying the types of patients who may benefit from them. Beyond their potential to improve left ventricular function or prevent sudden cardiac death, these devices can provide early warning signs of HF prodromal symptoms, such as increased ectopic activity, low percentages of biventricular pacing, or changes in patient activity.
The challenges of integrating data from these devices into healthcare systems are discussed in the chapter "Challenges to Implementing Wearable Data in Everyday Clinical Care."
Additionally, in response to your feedback, we have included a description of ECG and PPG signals in the context of HF detection, illustrating how these signals change when HF is present.
2.4. The author should summarize the number of publications, and the databases used in this study (e.g. pubmed, etc).
Thank you very much for your comment. The reference to the databases used has been included at the end of the introduction.
Reviewer 3 Report
Comments and Suggestions for Authors
I have two observations:
1) The Information in the following sections is missing:
Author Contributions: line 508
Funding: line 509
Institutional Review Board Statement: line 510
Informed Consent Statement: line 511
Data Availability Statement: line 512
Conflicts of Interest: line 513
2) The numbering of the articles in the bibliographic list must be corrected because it is duplicated.
Author Response
I have two observations:
1) The Information in the following sections is missing:
Author Contributions: line 508
Funding: line 509
Institutional Review Board Statement: line 510
Informed Consent Statement: line 511
Data Availability Statement: line 512
Conflicts of Interest: line 513
Thank you for the comment, the information has been added.
2) The numbering of the articles in the bibliographic list must be corrected because it is duplicated.
Thank you, the numbering of the bibliographic references has been changed